# SPATIALTHINKER: REINFORCING 3D REASONING IN MULTIMODAL LLMS VIA SPATIAL REWARDS

## ABSTRACT

Multimodal large language models (MLLMs) have achieved remarkable progress in vision–language tasks, but they continue to struggle with spatial understanding. Existing spatial MLLMs often rely on explicit 3D inputs or architecture-specific modifications, and remain constrained by large-scale datasets or sparse supervision. To address these limitations, we introduce SPATIALTHINKER, a 3D-aware MLLM trained with RL to integrate structured spatial grounding with multi-step reasoning. The model simulates human-like spatial perception by constructing a scene graph of task-relevant objects and spatial relations, and reasoning towards an answer via dense spatial rewards. SPATIALTHINKER consists of two key contributions: (1) a data synthesis pipeline that generates STVQA-7K, a high-quality spatial VQA dataset, and (2) online RL with a multi-objective dense spatial reward enforcing spatial grounding. SPATIALTHINKER-7B outperforms supervised fine-tuning and the sparse RL baseline on spatial understanding and real-world VQA benchmarks, nearly doubling the base-model gain compared to sparse RL, and surpassing GPT-4o. These results showcase the effectiveness of combining spatial supervision with reward-aligned reasoning in enabling robust 3D spatial understanding with limited data and advancing MLLMs towards human-level visual reasoning.

## 1 INTRODUCTION

Spatial reasoning is central to human intelligence, enabling us to perceive, localize, and manipulate objects in complex environments. This capability is crucial for embodied AI tasks such as robotic manipulation (Intelligence et al., 2025; Gao et al., 2023; Nasiriany et al., 2024), navigation (Huang et al., 2022), and augmented reality (Konenkov et al., 2024), where precise spatial awareness underpins interactive decision-making and makes spatial reasoning essential for real-world deployment (Driess et al., 2023; Team et al., 2025). While multimodal large language models (MLLMs) have advanced rapidly in vision-language tasks such as visual question answering (VQA), captioning and referring expression comprehension (Hurst et al., 2024; Lin et al., 2024; Deitke et al., 2025; Bai et al., 2025; Du et al., 2025; Liu et al., 2023; Google, 2025), they continue to struggle with spatial understanding tasks, especially in the 3D space (Chen et al., 2024a; Tong et al., 2024b; Kamath et al., 2023; Yang et al., 2025a; Tong et al., 2024a; Ma et al., 2024b), which requires capturing geometry, structure, and relations beyond 2D projections.

Existing approaches are often data-intensive, relying on either synthesizing massive question-answering datasets from 3D scene graphs (Chen et al., 2024a; Ma et al., 2025b; Daxberger et al., 2025; Cheng et al., 2024), training auxiliary spatial tokens or architectural changes (Hong et al., 2023b; Ma et al., 2025b), ingesting explicit 3D inputs like depth maps or point clouds (Hong et al., 2023c; Cheng et al., 2024; Cai et al., 2024), or more recently applying reinforcement learning (RL) with sparse rewards (Ma et al., 2025a; Wang & Ling, 2025; Xia et al., 2025; Xiao et al., 2025; Shen et al., 2025a; Zhu et al., 2025). This has led to models that are extremely data-hungry (e.g., SpatialVLM trained on 2B VQA samples (Chen et al., 2024a), SpatialLLM on 1M (Ma et al., 2025b), SpatialRGPT on 700k (Cheng et al., 2024)), or require architecture-specific modifications.

Recently, reinforcement learning with verifiable rewards (RLVR) has demonstrated superior generalization over supervised fine-tuning (SFT) by learning diverse reasoning strategies rather than static

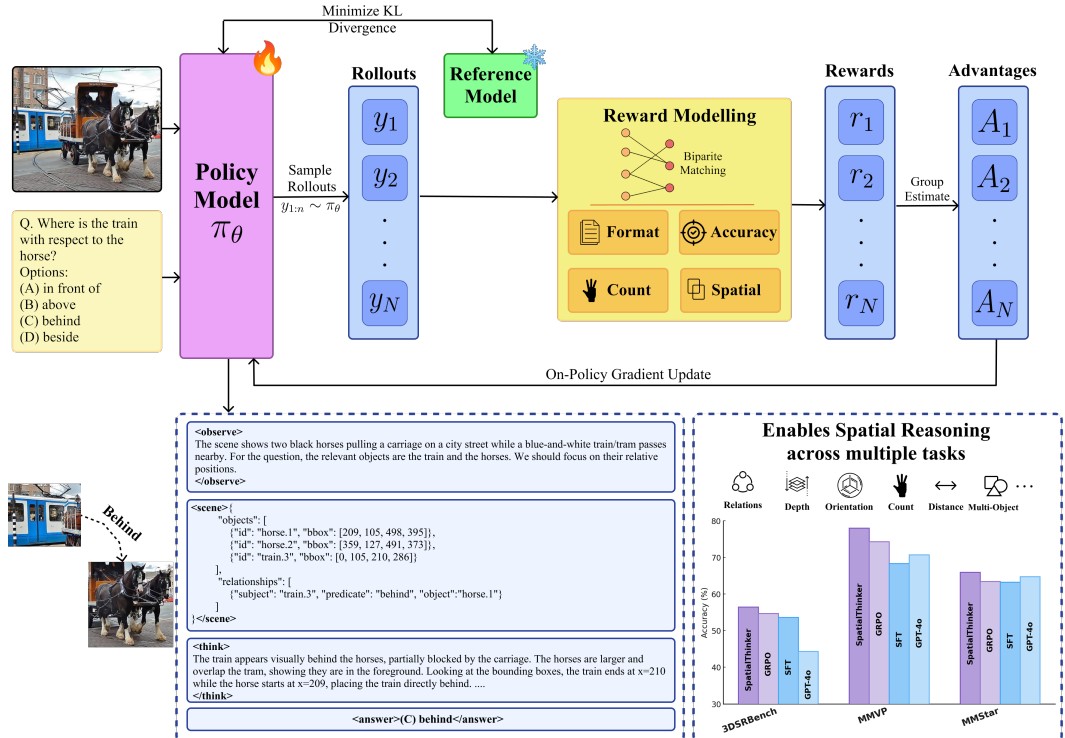

Figure 1: Method overview of SPATIALTHINKER. Our framework integrates structured scene-graph grounded reasoning with multi-objective dense RL to enhance 3D spatial understanding in multimodal large language models.

patterns (DeepSeek-AI et al., 2025; Shen et al., 2025b; Gandhi et al., 2025). However, existing RLVR approaches for visual spatial reasoning employ simple rewards focused on final correctness, providing insufficient guidance for visually-grounded reasoning (Shen et al., 2025a; Xiao et al., 2025; Ma et al., 2025a). We hypothesize that progress in this domain requires models to simulate grounded perception before reasoning, mirroring how humans mentally visualize regions of interest and relational layouts before making spatial judgments (Yang et al., 2016; Wu & Xie, 2023; Yang et al., 2025a). Scene graphs offer natural structure (Hildebrandt et al., 2020; Wald et al., 2020), but existing methods treat them as external pre-processing (Kim et al., 2024; Chen et al., 2023; Li et al., 2024c; Chen et al., 2025c; Li et al., 2025) rather than integrating them with end-to-end reasoning.

We introduce SPATIALTHINKER, a 3D-aware MLLM that integrates scene graph grounding with multi-step spatial reasoning through online policy RL. The model constructs question-focused scene subgraphs capturing objects, their relations, and localized coordinates, and reasons over these structured representations. The training leverages a multi-objective reward framework with lexicographic ordering: format rewards enforce structured reasoning; count penalties regulate regional focus; accuracy rewards prioritize correctness; and CIoU-based spatial rewards encourage precise localization when answers are correct. This design promotes human-like reasoning, following a process of *observe, localize, think, answer*.

By training on only 7K samples from our synthesized STVQA-7K dataset, SPATIALTHINKER-7B outperforms supervised fine-tuning (+6%) and conventional RL baselines (+3.2%) across twelve spatial understanding, real-world and generic VQA benchmarks, surpassing GPT-4o (+3.4% avg.) and Claude 3.5 Sonnet (+10.1% avg.) (Hurst et al., 2024; Anthropic, 2024), particularly a +12.1% gain over GPT-4o on 3DSRBench (Ma et al., 2024b). Notably, while vanilla RL with sparse rewards improves the base model by +4% average across all benchmarks, SPATIALTHINKER-7B trained with dense spatial rewards achieves +7.2% gains, almost doubling (×1.8) the benefit of RL training by providing richer learning signals. This demonstrates that models can learn effective spatial reasoning by discovering how to focus on regions of interest, construct mental scene graph representations, and accurately localize objects - all through online environmental feedback from dense rewards

that incentivize visually-grounded perception, rather than relying on data scale alone. The strong generalization for in-domain and out-of-domain tasks from minimal high-quality data validates that properly-guided RL surpasses static SFT patterns learned from much larger datasets (Chen et al., 2024a; Ma et al., 2024a). Our main contributions are:

- We propose SPATIALTHINKER, the first MLLM integrating scene graph-based grounding with online RL for spatial reasoning, achieving strong performance with only 7K training samples versus millions required by existing methods.

- We introduce STVQA-7K, a high-quality spatial VQA dataset grounded in scene graphs, enabling efficient training for spatial reasoning.

- We design a dense spatial rewead that prioritizes objectives in a fixed order (through lexicographic gating). This encourages interpretable, region-focused reasoning and prevents reward hacking.

- We evaluate the method on six spatial understanding, and six real-world VQA benchmarks demonstrating superior generalization performance.

## 2 PRELIMINARIES

**Scene Graph Generation.** A scene graph provides a structured representation of an image $I$ as a directed graph $G = (V, E)$. Each node $v_i \in V$ denotes an object with a category label $c_i$ and a 2D bounding box $b_i = (x_1, y_1, x_2, y_2)$; each edge $e_{ij} \in E$ is a relationship triplet $\langle v_i, r_{ij}, v_j \rangle$ consisting of subject $v_i$, predicate $r_{ij}$, and object $v_j$ that capture spatial or interactive relations (e.g., *left of*, *on*, *under*) (Hildebrandt et al., 2020; Wald et al., 2020). Classical SGG decomposes prediction into object detection and relation recognition (Carion et al., 2020; Cong et al., 2023), while open-vocabulary methods leverage language or vision priors to generalize beyond fixed ontologies (Chen et al., 2024b; Li et al., 2023). We refer to *question-focused scene subgraphs* as $G_q = (V_q, E_q) \subseteq G$ that retain only objects and relations relevant to a given query $q$.

**Reasoning in Multimodal Large Language Models.** Multimodal large language models (MLLMs) aim to solve reasoning tasks defined over a dataset $\mathcal{D}$ of multimodal instances $(\mathbf{x}_{\text{img}}, \mathbf{x}_{\text{text}}, \mathbf{y}^*)$, where $\mathbf{x}_{\text{img}}$ is a visual input, $\mathbf{x}_{\text{text}}$ is a natural language query, and $\mathbf{y}^*$ is a verifiable reasoning trajectory. We model the MLLM as an autoregressive policy $\pi_\theta$ that outputs a trajectory $\mathbf{y} = (s_1, \dots, s_T, a)$ consisting of reasoning steps $s_t$ and a final answer $a$. The policy factorizes as:

$$\pi_\theta(\mathbf{y} \mid \mathbf{x}_{\text{img}}, \mathbf{x}_{\text{text}}) = \left( \prod_{t=1}^{T} \pi_\theta(s_t \mid \mathbf{x}_{\text{img}}, \mathbf{x}_{\text{text}}, s_{<t}) \right) \cdot \pi_\theta(a \mid \mathbf{x}_{\text{img}}, \mathbf{x}_{\text{text}}, s_{\leq T}). \tag{1}$$

Supervised fine-tuning enables imitation of reference reasoning traces but often struggles with generalization. Reinforcement learning (RL) instead optimizes reasoning trajectories with explicit reward signals, improving robustness and task adherence (Gandhi et al., 2025; DeepSeek-AI et al., 2025; Huang et al., 2025). The RL objective is given by: $\max_\theta \ \mathbb{E}_{(\mathbf{x}_{\text{img}}, \mathbf{x}_{\text{text}}, \mathbf{y}^*) \sim \mathcal{D}, \ \mathbf{y} \sim \pi_\theta} [R(\mathbf{y})]$, where $R(\mathbf{y})$ evaluates the trajectory based on format adherence, object counting, answer correctness, and spatial localization.

## 3 SPATIALTHINKER: SPATIALLY-AWARE REASONING MLLMS

**Task Formulation** We cast spatial reasoning in MLLMs as the task of producing a visually grounded response $\mathbf{y}$ to a query $Q = \{\mathbf{x}_{\text{img}}, \mathbf{x}_{\text{text}}\}$. Unlike generic reasoning, our formulation explicitly requires constructing question-focused scene subgraphs $G_q$ and reasoning over objects, bounding boxes, and relations. The policy $\pi_\theta$ is trained on spatially grounded VQA samples from STVQA-7K 3.3 using our multi-objective spatial reward $R$ (Section 3.1), which enforces structural validity, count fidelity, answer accuracy, and precise spatial grounding.

### 3.1 MULTI-OBJECTIVE REWARD DESIGN

SPATIALTHINKER is trained with a fine-grained, multi-objective reward function that guides spatial reasoning via explicit visual grounding. Unlike prior RLVR methods that use sparse final-answer

rewards (Peng et al., 2025; Zhu et al., 2025; Shen et al., 2025b), our dense reward design combines lexicographic gating with four components—format, count, accuracy, and spatial rewards. We further discuss our reward design rationale in Appendix C.

**Format Reward.** We enforce a visually-grounded and structured reasoning template: `<observe>` for scene description, `<scene>` for regional scene graphs with objects, bounding boxes, and relations, `<think>` for explicit reasoning, and `<answer>` for the final output. Beyond tag presence, the format reward validates the JSON inside `<scene>`, ensuring (1) it is parseable, (2) each object includes required fields (ID and bounding box), and (3) all relations are valid subject–predicate–object triplets. This encourages sequential grounding: perceive → localize → reason → answer. The reward $R_f \in {0, 1}$ is weighted at $w_{\text{format}} = 0.1$.

**Accuracy Reward.** To prioritize task performance, we define the accuracy reward $R_a$ as a binary score based on exact string match between the model's predicted answer and the ground-truth answer, enabled by our multiple-choice format. This component carries the highest weight ($w_{\text{accuracy}} = 0.5$), directly incentivizing correct final predictions, while the other rewards shape how the model arrives at correct answers.

**Count Reward.** The count reward encourages the model to predict the appropriate number of objects and relations relevant to the query, penalizing both under- and over-generation based on the deviation between predicted and ground-truth counts:

$$R_{\text{count}} = w_{\text{count}} \cdot \left( 0.7 \cdot \max\left( 0, 1 - \frac{|N_{\text{obj}}^{\text{pred}} - N_{\text{obj}}^{\text{gt}}|}{\max(N_{\text{obj}}^{\text{gt}}, 1)} \right) + 0.3 \cdot \max\left( 0, 1 - \frac{|N_{\text{rel}}^{\text{pred}} - N_{\text{rel}}^{\text{gt}}|}{\max(N_{\text{rel}}^{\text{gt}}, 1)} \right) \right)$$

where $N^{\text{pred}}$ and $N^{\text{gt}}$ denote predicted and ground truth counts respectively, and $w_{\text{count}} = 0.2$ is the overall count reward weight. This guides the model to stay focused on question-relevant regions. Without it, we found the models tend to game the spatial reward by generating excessive objects and relations to maximize random matches—a form of reward hacking.

**Spatial Reward.** To supervise object localization, we compute the spatial reward only when the final answer is correct. Predicted and ground-truth objects are matched using the Hungarian algorithm for bipartite matching with a cost function that combines Complete IoU (CIoU) and semantic similarity:

$$C(o_i^{\text{pred}}, o_j^{\text{gt}}) = \lambda_{\text{spatial}}(1 - \text{IoU}(b_i, b_j)) + \lambda_{\text{semantic}}(1 - \text{sim}(l_i, l_j)),$$

where $b$ and $l$ denote bounding boxes and labels, respectively, $\lambda_{spatial} = 1.0$, and $\lambda_{semantic} = 2.0$. The reward is then computed as the average CIoU across matched pairs: $R_{\text{spatial}} = w_{\text{spatial}} \cdot \left( \frac{1}{|\mathcal{M}|} \sum_{(i,j) \in \mathcal{M}} \text{CIoU}(b_i^{\text{pred}}, b_j^{\text{gt}}) \right)$, where $w_{\text{spatial}} = 0.2$. CIoU offers dense supervision over IoU, even for non-overlapping boxes by incorporating distance and aspect ratio terms Zheng et al. (2020).

**Lexicographic Gating.** To avoid reward gaming across objectives, we apply lexicographic ordering with conditional gating Skalse et al. (2022), prioritizing format $\succ$ {count, accuracy} $\succ$ spatial. The model must first satisfy formatting, then jointly optimize count and accuracy, and receives spatial reward only when the answer is correct. This ensures spatial grounding reinforces valid reasoning. Without accuracy gating, we observe that models overfit to spatial localization while sacrificing task correctness. The final reward is computed as the following with $\mathbb{I}[\cdot]$ as the indicator function:

$$R_{\text{total}} = \mathbb{I}[R_{\text{format}} = 1] \cdot (w_{\text{format}}R_f + w_{\text{count}}R_c + w_{\text{accuracy}}R_a + \mathbb{I}[R_{\text{accuracy}} = 1]w_{\text{spatial}}R_s)$$

## 3.2 ONLINE RL POLICY OPTIMIZATION

To train SPATIALTHINKER with dense, lexicographically gated rewards, we adopt Group-Relative Policy Optimization (GRPO) DeepSeek-AI et al. (2025); Shao et al. (2024), an online RL method that avoids critic networks by estimating advantages through intra-group comparisons. Given an input $\mathbf{x}$, we sample $N$ trajectories $\{y^{(1)}, \ldots, y^{(N)}\}$ from the current policy $\pi_{\theta_{\text{old}}}$. Each response is scored via our dense spatial reward function (Section 3.1), and advantages are computed using group-normalized scores: $A^{(i)} = \frac{r^{(i)} - \mu}{\sigma + \varepsilon}$, where $\mu$ and $\sigma$ are the group mean and standard deviation, and $\varepsilon = 10^{-6}$. We then update the policy using a PPO-style clipped loss with KL regularization:

$$\mathcal{L}_{\text{RL}}(\theta) = -\frac{1}{G} \sum_{i=1}^{G} \frac{1}{|y^{(i)}|} \sum_{t=1}^{|y^{(i)}|} \left[ \min\left( r^{i,t} A^{(i)}, \text{clip}(r^{i,t}, 1 - \epsilon_l, 1 + \epsilon_h) A^{(i)} \right) - \beta D_{\text{KL}}^{i,t} \right],$$

where $r^{i,t} = \frac{\pi_\theta(y_t^{(i)}|\mathbf{x},y_{<t}^{(i)})}{\pi_{\theta_{\text{old}}}(y_t^{(i)}|\mathbf{x},y_{<t}^{(i)})}$ is the importance ratio between new and old policies, and $D_{\text{KL}}^{i,t}$ is the token-level KL divergence against a reference model. We set $\epsilon_l = 0.2$, $\epsilon_h = 0.3$, and $\beta = 10^{-2}$. This objective balances learning from dense spatial rewards while constraining policy divergence to ensure stability and generalization.

## 3.3 STVQA-7K: DATASET CONSTRUCTION

To facilitate reward-aligned spatial reasoning, we construct STVQA-7K, a synthetic visual question answering (VQA) dataset built from human-annotated scene graphs in Visual Genome Krishna et al. (2017). STVQA-7K comprises 7,587 spatially grounded multiple-choice VQA pairs spanning both 2D and 3D spatial understanding, covering nine core reasoning types including relations, size, orientation, distance, depth, reach, location, count, and existence. We augment the original VG150 predicate set with 34 additional spatial relations—covering distance (e.g., near, far), size (e.g., bigger, taller), orientation (e.g., facing away), and containment (e.g., inside, beneath)—to enrich the relational vocabulary beyond the standard 50 predicates. Each QA pair is generated from a scene graph using Claude Sonnet 4 Anthropic (2025), and rated by difficulty and quality. We apply a consistency-based filtering pipeline using GPT-4o Hurst et al. (2024) to ensure semantic correctness via pass@2 agreement. From an initial pool of 56,224 questions, we retain the top 7.5K high-quality samples based on rating, difficulty, and verification. To enable region-specific reasoning, we extract relevant

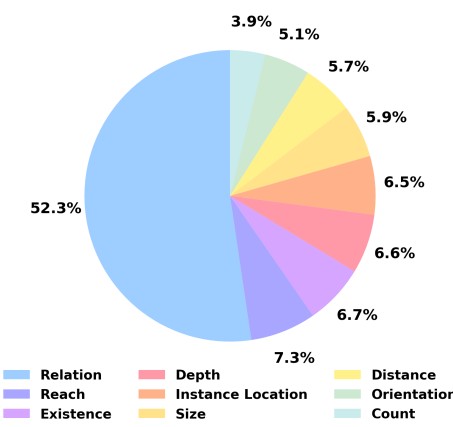

Figure 2: Distribution of QA types in STVQA-7K. The dataset spans a diverse range of spatial reasoning skills, covering spatial relations, localization, existence, reach, depth, distance, size, count and orientation.

objects and relations per question via lemmatized keyword matching, constructing question-aligned scene subgraphs as localized supervision. This localized supervision helps the model learn where to focus within complex scenes. Bounding box coordinates are retained in absolute pixel space to preserve real-world scale for CIoU-based reward training. Importantly, our pipeline is scalable and can be extended to generate up to ∼108K samples, the maximum supported by Visual Genome, enabling future large-scale post-training or RL fine-tuning. Figure 2 shows the distribution of QA categories. Full dataset details and examples are provided in Appendix A.

## 3.4 TRAINING DETAILS

We build SPATIALTHINKER upon two strong open-source multimodal base models: Qwen2.5-VL-3B and Qwen2.5-VL-7B (Bai et al., 2025), using them as backbones for policy optimization with RL. No SFT is performed prior to RL training on our STVQA-7K dataset (Section 3.3). We employ GRPO (Shao et al., 2024) as the advantage estimator as described in Section 3.2, using a rollout size of 8 samples per query and a sampling temperature of $1.0$. The models are trained with a maximum context length of 16,384 tokens. The rollout batch size is set to 512, and the global batch size is 128. We train for 75 training steps i.e., 5 training episodes) on $4 \times$ NVIDIA H100 80GB GPUs. Training time totals $\sim 13$ hours for the 3B model and $\sim 15$ hours for the 7B model.

The models are trained on high-resolution image inputs ranging from $512 \times 512$ to $2048 \times 2048$ pixels, to preserve fine-grained spatial information. All model parameters, including the vision encoder, are updated during training. We use the AdamW optimizer with `bf16` precision, a learning rate of $1 \times 10^{-6}$, and a weight decay of $1 \times 10^{-2}$. The KL penalty coefficient is set to $10^{-2}$ (Appendix D). STVQA-7K is partitioned with a 90/10 train–validation split. Further details on prompts, SFT, and RL training setups, are provided in Appendices B.3, B.4, and B.5, respectively. Finally, Section B.5.1 illustrates how each reward component improves steadily under our multi-objective spatial reward, reflecting stable and interpretable learning dynamics.

| Model | 3DSRBench | CV-Bench | | Avg. | BLINK$_{val}$ | | Avg. |
|---|---|---|---|---|---|---|---|
| | | 2D | 3D | | Spatial Relation | Relative Depth | |
| *Proprietary Models* | | | | | | | |
| GPT-4o | 44.3 | 75.8 | **83.0** | **79.4** | 82.5 | **78.2** | **80.4** |
| Claude 3.5 Sonnet | 48.2 | 60.2 | 71.5 | 65.9 | 58.7 | 67.7 | 63.2 |
| *Open-Source General MLLMs* | | | | | | | |
| Qwen2.5-VL-3B | 44.0 | 59.9 | 60.2 | 60.0 | 66.4 | 54.0 | 60.2 |
| Qwen2.5-VL-7B | 48.4 | 69.1 | 68.0 | 68.6 | 84.0 | 52.4 | 68.2 |
| VLAA-Thinker-Qwen2.5-VL-7B | 52.2 | 60.8 | 60.3 | 60.6 | 81.2 | 71.0 | 76.1 |
| LLaVA-NeXT-8B | 48.4 | 62.2 | 65.3 | 63.8 | - | - | - |
| Cambrian-1-8B | 42.2 | 72.3 | 72.0 | 72.2 | 69.9 | 73.4 | 71.7 |
| *Open-Source Spatial MLLMs* | | | | | | | |
| RoboPoint-13B | - | - | 61.2 | - | 60.8 | 61.3 | 61.1 |
| SpatialBot-3B | 41.1 | - | 69.1 | - | 67.8 | 67.7 | 67.8 |
| SpaceLLaVA-13B | 42.0 | - | 68.5 | - | 72.7 | 62.9 | 67.8 |
| Spatial-RGPT-7B w/ depth | 48.4 | - | 60.7 | - | 65.7 | 82.3 | 74.0 |
| SpaceThinker | 51.1 | 65.1 | 65.9 | 65.5 | 73.4 | 59.9 | 66.7 |
| SpaceOm | 52.2 | 72.1 | 69.3 | 70.7 | 81.1 | 65.3 | 73.2 |
| *Method Comparison (Trained on SpatialThinkerVQA)* | | | | | | | |
| Qwen2.5-VL-3B + SFT | 50.8 | 53.9 | 68.4 | 61.1 | 65.0 | 66.9 | 66.0 |
| Qwen2.5-VL-3B + Vanilla GRPO | 50.1 | 70.6 | 66.6 | 68.6 | 73.4 | 55.6 | 64.5 |
| **SpatialThinker-3B (Ours)** | 52.9 | 71.0 | 76.3 | 73.6 | 81.8 | 66.9 | 74.4 |
| Qwen2.5-VL-7B + SFT | 53.6 | 56.1 | 71.3 | 63.7 | 75.5 | 64.5 | 70.0 |
| Qwen2.5-VL-7B + Vanilla GRPO | 54.7 | 68.9 | 76.5 | 72.7 | 80.4 | 75.0 | 77.7 |
| **SpatialThinker-7B (Ours)** | **56.4** | **77.7** | 78.7 | 78.2 | **86.0** | 72.6 | 79.3 |

Table 1: Performance over 2D & 3D Spatial Understanding Benchmarks across different model types. Top-1 & Top-2 accuracies are represented using **bold text**, and underlines.

## 4 EXPERIMENTS

We evaluate SPATIALTHINKER across 12 diverse spatial understanding and real-world VQA benchmarks, encompassing both 2D and 3D reasoning tasks. Our experiments are guided by two core questions: (Q1) Does our spatial VQA generation pipeline, combined with dense reward RL, improve general spatial reasoning in MLLMs? (Q2) Can MLLMs learn strong spatial capabilities from just 7K synthetic training samples, and how does this compare to models trained on orders-of-magnitude more data?

**Benchmarks.** We evaluate on six core spatial benchmarks: CV-Bench 2D and 3D (Tong et al., 2024a), BLINK Spatial Relations and Relative Depth (Fu et al., 2024), 3DSRBench (Ma et al., 2024b), MMVP (Tong et al., 2024b), SpatialBench (Cai et al., 2024), and SpatialReasonerEval (Ma et al., 2025a), covering relation understanding, depth, distance, counting, size, and egocentric 3D reasoning. To test generalization in real-world, embodied, and generalist VQA contexts, we use VStarBench (Wu & Xie, 2023), RealWorldQA (xAI, 2024), MME-RealWorld (Zhang et al., 2024), RoboSpatial-Home (Song et al., 2025) (Configuration and Compatibility only), MM-Star (Chen et al., 2024c), and HallusionBench (Guan et al., 2023).

**Baselines.** We compare against proprietary MLLMs including GPT-4o (GPT-4O-0513) (Hurst et al., 2024) and Claude 3.5 Sonnet (CLAUDE-3.5-SONNET-0620) (Anthropic, 2025), open-source generalist models like Qwen2.5-VL (Bai et al., 2025), LLaVA-NeXT (Li et al., 2024b), Cambrian-1 (Tong et al., 2024a), and VLAA-Thinker (Chen et al., 2025a), and spatially-tuned open-source MLLMs such as SpaceLLaVA (AI & Mayorquin, 2025a; Chen et al., 2024a), SpatialRGPT (Cheng et al., 2024), RoboPoint (Yuan et al., 2024), SpaceThinker (AI & Mayorquin, 2025c), SpaceOm (AI & Mayorquin, 2025b), SpatialReasoner (Ma et al., 2025a), and SpatialBot (Cai et al., 2024). In addition, we evaluate ablations on variants of our model trained with the STVQA-7K dataset: a supervised fine-tuning (SFT) baseline, and a sparse-reward RL baseline that optimizes only format and accuracy rewards, each weighted equally at 0.5, , to isolate the effect of our dense spatial reward.

**Evaluation Setting.** All models are evaluated in a zero-shot setting using greedy decoding (temperature = 0.0). Models default prompting format is used where applicable (e.g., for VLAA-Thinker, SpaceOm, SpaceThinker). SpatialRGPT is evaluated with depth inputs; all other models use RGB. Accuracy is the primary evaluation metric. Our evaluation pipeline extends OpenVLThinker (Deng et al., 2025) to support new benchmarks and formats. *Full benchmark descriptions, baseline details, and additional implementation specifics are provided in Appendix B.*

## 4.1 RESULTS

We evaluate SpatialThinker across six spatial reasoning and six generalist VQA benchmarks to assess its effectiveness in learning spatial understanding and real-world VQA from limited training data through dense reward supervision.

**Performance across Spatial Benchmarks.** We evaluate SPATIALTHINKER across six spatial reasoning benchmarks that collectively span 2D relational understanding, 3D spatial alignment, counting, depth ordering, and distance comparison. As shown in Tables 1 and 2, SPATIALTHINKER-7B achieves strong and consistent performance across all spatial tasks. On CV-Bench, the model attains an average accuracy of 78.2% across 2D and 3D tasks, nearing GPT-4o's 79.4% while outperforming all other open-source models, and Claude 3.5 Sonnet. On the challenging 3DSRBench, which requires orientation and multi-object reasoning, it achieves 56.4%, surpassing GPT-4o by +12%. On BLINK's spatial relation and relative depth tasks, it achieves 86.0% and 72.6%, respectively,

| Model | MMVP | SpatialReasonerEval | SpatialBench |
|---|---|---|---|
| *Proprietary and Open-Source MLLMs* | | | |
| GPT-4o | 70.7 | **85.8** | **67.0** |
| Claude 3.5 Sonnet | 71.3 | 84.1 | 63.2 |
| Qwen2.5-VL-3B | 67.0 | 68.0 | 49.9 |
| Qwen2.5-VL-7B | 72.3 | 70.6 | 62.5 |
| VLAA-Thinker-7B | 75.3 | 61.2 | 66.2 |
| SpaceThinker | 63.0 | 69.6 | 57.9 |
| SpaceOm | 66.3 | 68.9 | 58.6 |
| SpatialReasoner | 64.0 | 76.4 | 59.2 |
| *Method Comparison (Trained on SpatialThinkerVQA)* | | | |
| Qwen2.5-VL-3B + SFT | 62.7 | 67.5 | 56.3 |
| Qwen2.5-VL-3B + Vanilla GRPO | 68.3 | 69.3 | 56.9 |
| **SpatialThinker-3B (Ours)** | 69.0 | 76.5 | 61.5 |
| Qwen2.5-VL-7B + SFT | 68.3 | 70.8 | 63.5 |
| Qwen2.5-VL-7B + Vanilla GRPO | 74.3 | 79.6 | 64.2 |
| **SpatialThinker-7B (Ours)** | **78.0** | 82.7 | 66.4 |

Table 2: Performance on additional spatial benchmarks. Top-1 & Top-2 accuracies are represented using **bold text**, and underlines.

yielding a 79.3% average—closely matching GPT-4o (80.4%) and outperforming other spatial MLLMs like Spatial-RGPT-7B (74.0%), which uses depth inputs and 700K training samples. On SpatialBench, our model reaches 66.4%, approaching GPT-4o's 67.0%.

Despite being trained on just 7K synthetic samples and using only RGB inputs, SPATIALTHINKER-7B consistently outperforms open-source baselines, including VLAA-Thinker-7B, Cambrian-1-8B, Spatial-RGPT, SpaceLLaVA, and RoboPoint-13B, all of which are trained on orders of magnitude more data. Notably, it exceeds specialized spatial models as well: on CV-Bench 3D, it outperforms SpaceLLaVA-13B (78.7% vs. 68.5%), and on BLINK tasks, it surpasses Spatial-RGPT-7B by +5.3%, and SpatialBot by +11.5% despite their reliance on depth information. Further, SPATIALTHINKER-7B outperforms all models on MMVP, and all open-source baselines on SpatialReasonerEval that measures 3D spatial understanding tasks like depth and distance. These results highlight the effectiveness of our dense reward design in enabling generalizable spatial reasoning without the need for explicit geometric inputs or large-scale pretraining.

| Model | MM-Star | VStarBench | RealWorldQA | MME-RealWorld-Lite | RoboSpatial-Home | HallusionBench |
|---|---|---|---|---|---|---|
| *Proprietary and Open-Source MLLMs* | | | | | | |
| GPT-4o | 64.7 | 66.0 | **75.4** | **51.6** | 68.4 | 55.0 |
| Claude 3.5 Sonnet | 65.1 | 51.8 | 60.1 | 45.2 | 57.0 | 55.5 |
| Qwen2.5-VL-3B | 55.9 | 74.9 | 58.2 | 41.9 | 58.7 | 46.3 |
| Qwen2.5-VL-7B | 63.9 | 75.9 | 68.4 | 44.1 | 70.6 | 52.9 |
| VLAA-Thinker-7B | 63.8 | 58.1 | 66.4 | 44.6 | 68.9 | **68.9** |
| SpaceThinker | 54.5 | 56.5 | 61.6 | - | 52.6 | 65.4 |
| SpaceOm | 57.7 | 56.5 | 53.3 | - | 68.9 | 62.9 |
| *Method Comparison (Trained on SpatialThinkerVQA)* | | | | | | |
| Qwen2.5-VL-3B + SFT | 53.9 | 73.3 | 64.8 | 43.0 | 69.8 | 58.9 |
| Qwen2.5-VL-3B + Vanilla GRPO | 56.7 | 74.3 | 64.4 | 46.7 | 64.0 | 59.0 |
| **SpatialThinker-3B (Ours)** | 57.6 | 78.0 | 66.3 | 46.5 | 70.6 | 62.5 |
| Qwen2.5-VL-7B + SFT | 63.2 | 78.0 | 65.4 | 47.4 | 72.4 | 66.2 |
| Qwen2.5-VL-7B + Vanilla GRPO | 63.4 | 73.9 | 66.6 | 46.3 | 76.2 | 60.7 |
| **SpatialThinker-7B (Ours)** | **65.9** | **81.7** | 69.2 | 48.3 | **76.3** | 66.4 |

Table 3: Performance on VQA and Real-World benchmarks. Top-1 & Top-2 accuracies are represented using **bold text**, and underlines.

**Performance across Real-World and General VQA Benchmarks.** We further assess our model's generalization to real-world visual question answering using six diverse benchmarks: MM-Star, RealWorldQA, VStarBench, MME-RealWorld-Lite, RoboSpatial-Home, and HallusionBench (Table 3). SPATIALTHINKER-7B achieves the highest overall performance across these datasets. It obtains 65.9% on MM-Star, 81.7% on VStarBench, and 76.3% on RoboSpatial-Home, surpassing all open-source and proprietary baselines. It also performs competitively on hallucination-sensitive and

real-world benchmarks, scoring 66.4% on HallusionBench, 69.2% on RealWorldQA, and 48.3% on MME-RealWorld-Lite benchmarks.

These results show that training with dense spatial rewards generalizes beyond synthetic benchmarks to real-world settings. Gains on MM-Star, RoboSpatial-Home, and VStarBench highlight the benefit of structured scene grounding, even with a small synthetic training set. Compared to generalist and open-source spatial MLLM baselines, SPATIALTHINKER delivers greater robustness, fewer hallucinations, and higher task fidelity, reinforcing our hypothesis that spatial grounding via reward optimization not only improves spatial reasoning but also enhances visual understanding in the wild.

**RL Training with Dense Rewards Enables Superior Generalization.** To isolate the contributions of our multi-objective spatial reward design, we compare against two ablation variants: supervised fine-tuning (SFT) and reinforcement learning with sparse rewards using only format and answer accuracy. As shown in Table 4, SPATIALTHINKER-7B achieves an average accuracy of 71.2% across all 12 benchmarks—exceeding the SFT baseline by +6.0% and the sparse GRPO variant by +3.2%. These gains are consistent across the 3B variant as well, where SPATIALTHINKER-3B outperforms its SFT and GRPO counterparts by +5.5% and +4.1% average

| Model | Avg. Acc. (12) | $\Delta_{\text{Base}}$ | $\Delta_{\text{GPT-4o}}$ | $\Delta_{\text{Claude 3.5 Sonnet}}$ |
|---|---|---|---|---|
| *Proprietary and Base MLLMs* | | | | |
| GPT-4o | 67.8 | - | - | - |
| Claude 3.5 Sonnet | 61.1 | - | - | - |
| Qwen2.5-VL-3B | 57.3 | - | - | - |
| Qwen2.5-VL-7B | 64.0 | - | - | - |
| *Method Comparison (Trained on SpatialThinkerVQA)* | | | | |
| Qwen2.5-VL-3B + SFT | 60.8 | +3.5 | -7.0 | -0.3 |
| Qwen2.5-VL-3B + Vanilla GRPO | 62.2 | +4.9 | -5.6 | +1.1 |
| **SpatialThinker-3B (Ours)** | 66.3 | +9.0 | -1.5 | +5.2 |
| Qwen2.5-VL-7B + SFT | 65.2 | +1.2 | -2.6 | +4.1 |
| Qwen2.5-VL-7B + Vanilla GRPO | 68.0 | +4.0 | +0.2 | +6.9 |
| **SpatialThinker-7B (Ours)** | **71.2** | **+7.2** | **+3.4** | **+10.1** |

Table 4: Average accuracy across all 12 benchmarks with relative improvements ($\Delta$). *SpatialThinker models consistently outperform SFT and vanilla GRPO, with SpatialThinker-7B surpassing GPT-4o by +3.4 points and Claude 3.5 Sonnet by +10.1 points.*

gains, respectively. Notably, even vanilla GRPO provides modest improvements over the base model (+4.0 for 7B, +4.9 for 3B), but our dense spatial reward nearly doubles $\times 1.8$ this gain (+7.2% for 7B, +9.0% for 3B), underscoring the complementary learning signal provided by count and spatial objectives.

Beyond aggregate accuracy, lexicographic reward gating stabilizes training by enforcing format and answer correctness before applying spatial rewards. This encourages structured task completion prior to spatial grounding, resulting in steady and interpretable reward curves during training (Section B.5.1). Overall, these results affirm that structured reinforcement learning with dense spatial supervision significantly enhances the capabilities of multimodal LLMs, even in low-data regimes.

**Out-of-Distribution Generalization: Dense Rewards Enable Stronger Transfer.** While both SFT and sparse-reward GRPO improve spatial reasoning over base models, their ability to generalize to out-of-distribution (OOD) real-world tasks is limited, when compared to SPATIAL-THINKER models. As shown in Table 5, sparse-reward GRPO provides large spatial gains (+4.3% for 3B, +4.7% for 7B), but offers only marginal improvements on real-world benchmarks (+6.0 and +2.7 respectively)—nearly matching or

| Model Variant | Spatial VQA $\Delta_{\text{Base}}$ | Real-World VQA $\Delta_{\text{Base}}$ |
|---|---|---|
| Qwen2.5-VL-3B + SFT | +2.3 | +5.9 |
| Qwen2.5-VL-3B + GRPO | +4.3 | +6.0 |
| **SpatialThinker-3B** | **+9.3** | **+8.5** |
| Qwen2.5-VL-7B + SFT | +0.3 | +2.9 |
| Qwen2.5-VL-7B + GRPO | +4.7 | +2.7 |
| **SpatialThinker-7B** | **+8.3** | **+5.2** |

Table 5: Average accuracy gains ($\Delta$) over respective base models on (6) spatial and (6) real-world VQA (OOD) benchmarks.

underperforming SFT (+5.9% for 3B, +2.9% for 7B). In contrast, SPATIALTHINKER, trained with dense spatial and count rewards, achieves significantly stronger OOD generalization: +8.5 for 3B and +5.2 for 7B, outperforming all baselines at both scales. Notably, SPATIALTHINKER-7B provides nearly double the real-world VQA benchmarks gains compared to sparse-reward GRPO (+5.2% vs. +2.7%), highlighting the robustness of our dense reward framework. The combination of structured reasoning formats and lexicographically gated dense rewards encourages models to internalize spatial priors and compositional patterns that transfer effectively to out-of-distribution tasks, even without explicit domain-specific supervision. Appendix E further demonstrates generalization to abstract reasoning tasks.

## 5 RELATED WORK

**3D Spatial Reasoning in MLLMs.** While MLLMs have advanced core visual tasks (Hurst et al., 2024; Lin et al., 2024; Deitke et al., 2025; Bai et al., 2025; Du et al., 2025; Li et al., 2024b), their spatial reasoning abilities remain limited (Mirzaee et al., 2021; Tong et al., 2024b; Kamath et al., 2023; Yamada et al., 2023; Li et al., 2024a; Yang et al., 2025a; Ma et al., 2024b), partly due to datasets focused more on perception than relational grounding (Hudson & Manning, 2019). To address this, recent work integrates 3D signals via point clouds or multi-view reconstructions (Hong et al., 2023c;a), or world models with physical priors (Wang et al., 2023; 2024). Large-scale efforts like SpatialVLM (Chen et al., 2024a), SpatialPIN (Ma et al., 2024a), SpatialBot (Cai et al., 2024), and SpatialRGPT (Cheng et al., 2024) use hundred thousand to millions of 3D-augmented samples or RGB-D scene graphs. Others like MM-Spatial (Daxberger et al., 2025), SpatialLLM (Ma et al., 2025b), and SpaRE (Ogezi & Shi, 2025) similarly scale synthetic or reconstructed 3D data. However, these methods are often data-heavy, depend on specialized inputs, or fall short on structured relational modeling. In contrast, SPATIALTHINKER attains robust relational, and regional reasoning using just 7K structured QA samples trained with RL with dense spatial rewards.

**Structured Visual Grounding in MLLMs.** Scene graphs offer structured object–relation representations and have long supported visual reasoning (Hildebrandt et al., 2020; Wald et al., 2020; Gu et al., 2023). Classical Scene Graph Generation (SGG) relies on detection–relation pipelines (Carion et al., 2020; Cong et al., 2023), but struggles with multi-role and open-vocabulary generalization. Recent LLM-based methods like LLM4SGG and GPT4SGG extract structured graphs from captions (Kim et al., 2024; Chen et al., 2023), while open-vocabulary SGG approaches use MLLMs to generalize beyond fixed ontologies (Chen et al., 2024b; Li et al., 2023). RL-trained models like R1-SGG and Relation-R1 directly generate scene graphs via dense structural or cognitive rewards (Chen et al., 2025c; Li et al., 2025), emphasizing the value of structured supervision. In parallel, region-aware MLLMs including KOSMOS-2 (Peng et al., 2023), Ferret (You et al., 2023), and GLaMM (Rasheed et al., 2024), enhance spatial grounding via bounding boxes and region-text alignment. SPATIAL-THINKER extends these ideas by grounding reasoning in scene subgraphs focused on the question's region of interest, combining structured understanding with interpretable, reward-guided spatial reasoning.

**Multimodal Reinforcement Learning.** Reinforcement learning (RL) has been increasingly applied to enhance reasoning in MLLMs, extending chain-of-thought prompting (Wei et al., 2022) with verifiable rewards across tasks like math reasoning (Yang et al., 2025b; Meng et al., 2025), classification and grounding (Liu et al., 2025b), semantic segmentation (Liu et al., 2025a), regional understanding (Shen et al., 2025a), and open-vocabulary detection or referring expression comprehension (Shen et al., 2025b; Pinto et al., 2023). Spatial RL has also emerged, with SVQA-R1 using view-consistency rewards (Wang & Ling, 2025) and SpatialReasoner introducing coordinate-aware supervision (Shen et al., 2025b; Ma et al., 2025a). However, most prior methods rely on sparse signals like final accuracy or coarse location cues, offering limited support for fine-grained spatial reasoning. In contrast, SPATIALTHINKER introduces a dense, multi-objective reward framework encompassing regional subgraph construction, object localization, relational grounding, object counting, and final correctness. It first predicts structured scene representations, then reasons over them for detailed and interpretable spatial inference.

## 6 CONCLUSION

We introduced SPATIALTHINKER, a 3D-aware MLLM that achieves strong spatial reasoning by combining scene graph grounding with dense spatial rewards through RL. Trained on just 7K samples, it surpasses proprietary and open-sourced MLLMs on spatial, real-world, and generic VQA benchmarks while outperforming models trained on orders of magnitude more data, specifically for spatial understanding. Dense spatial rewards nearly double the gains of standard RL via GRPO, underscoring the value of rich supervision signals. While our approach relies on explicit scene graphs, future work could explore implicit spatial reasoning within latent tokens. Additional directions include extending our reward framework to spatiotemporal reasoning, real-world tasks like web navigation, and developing unified multi-objective policies covering diverse visual tasks.

## 7 REPRODUCIBILITY STATEMENT

To ensure the reproducibility of our results, we provide comprehensive details of our experimental setup in Section 3.4. Our dense reward design for RL training is described in Section 3.1, with further details on RL training, inference prompts, and baseline SFT training elaborated in Appendix B. Section Appendix A covers the dataset construction process and we will open-source our dataset on HuggingFace post the reviews. Our full code for training SpatialThinker models, dataset generation pipeline, and evaluation scripts, along with our STVQA-7K dataset and SpatialThinker 3B and 7B model checkpoints will be open-sourced after review on GitHub and HuggingFace.

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

APPENDIX

## A  STVQA-7K: DATASET CONSTRUCTION

High-quality spatial VQA datasets remain scarce, as most existing benchmarks either lack grounded scene-graph annotations (i.e., explicit spatial coordinates for objects and relations) or fail to comprehensively cover both 2D and 3D spatial reasoning categories. Visual Genome (Krishna et al., 2017) provides dense, human-annotated scene graphs that support strict grounding of both question generation and answer verification within a unified representational framework. Using Visual Genome, we synthetically constructed a spatial visual question answering dataset called SPATIALTHINKER Visual Question Answering dataset i.e., STVQA-7K comprising 7,587 samples, fully grounded in human-annotated scene graphs (Krishna et al., 2017), which we employed for post-training the SPATIALTHINKER models. Importantly, our pipeline is scalable and can be extended to generate up to 108K samples, the maximum supported by Visual Genome, enabling future large-scale post-training or RL fine-tuning.

The original VG150 predicate set is limited to 50 relations, missing several important categories such as positional relations (e.g., left, right, beside), distance-based relations (e.g., near, far, next to), comparative size (e.g., smaller, taller, bigger), orientation (e.g., facing towards/away), and containment (e.g., inside, beneath). To address this gap, we extended the scene graph relation space with an additional 34 predicates, ensuring richer spatial coverage in both 2D and 3D reasoning. Bounding box coordinates are retained in absolute pixel space, rather than normalized values, to preserve real-world scale and spatial alignment, to enable both improved spatial reasoning and effective use of CIoU-based supervision during reward optimization. The dataset construction pipeline proceeds in three stages: (1) synthetic question generation from ground-truth scene graphs, (2) automated quality filtering with external verification, and (3) scene graph adaptation for regional alignment with individual questions.

**Synthetic Question Generation.**   Visual Genome scene graphs serve as our foundational ground truth, providing object categories, bounding boxes, and relational triplets for over 150,000 images. We synthetically generate question-answer pairs for a given scene graph data using Claude Sonnet 4 (Anthropic, 2025), synthesizing multiple-choice questions based on the salient objects and meaningful spatial relations explicitly present in each graph. Each question-answer pair is accompanied with a rating generated out of 10 and the difficulty level. Our question generation encompasses nine distinct spatial reasoning categories: spatial relations (above, behind, near, etc.), physical reach and interaction (holding, touching), comparative size, orientation from specific viewpoints, instance location within image frames, depth ordering relative to the camera, distance comparisons to reference objects, object counting, and existence verification. This comprehensive taxonomy spans both 2D and 3D spatial understanding, providing a broad coverage of visual-spatial reasoning capabilities. To promote robust perception, we also include questions involving objects that are partially visible or occluded in the scene, encouraging the model to reason about spatial arrangements and fine-grained details. For each question, we generate a rating out of 10.

**Quality Filtering and Validation.**   To ensure semantic correctness at scale, we implement a consistency-based verification procedure using GPT-4o (Hurst et al., 2024) as an external validation model. For each generated question-answer pair, we assess agreement between the external model and our synthetic ground truth label using a pass@2 criterion. Questions that fail this initial consistency check undergo additional evaluation with two supplementary model responses. Items for which all four collected responses disagree with the generated label are discarded as potentially incorrect or ambiguous. This filtering process begins with 56,224 initially generated questions by Claude Sonnet 4 (Anthropic, 2025). We select the 10,000 highest-rated samples based on the questions complexity and rating towards its contribution to enhance spatial intelligence as judged by Claude Sonnet 4. Following consistency filtering, we retain 6,895 training samples and 692 validation samples ( 75%), indicating high label reliability. The final set consists of  50% samples from the relation category, and the remaining 50% distributed across the eight other categories. To prevent positional bias, answers are uniformly distributed across options A, B, C, and D. Figure Figure 2 illustrates the distribution of QA types in STVQA-7K, highlighting the emphasis on spatial relations while maintaining balanced coverage across the remaining reasoning categories. Representative examples of generated QA pairs

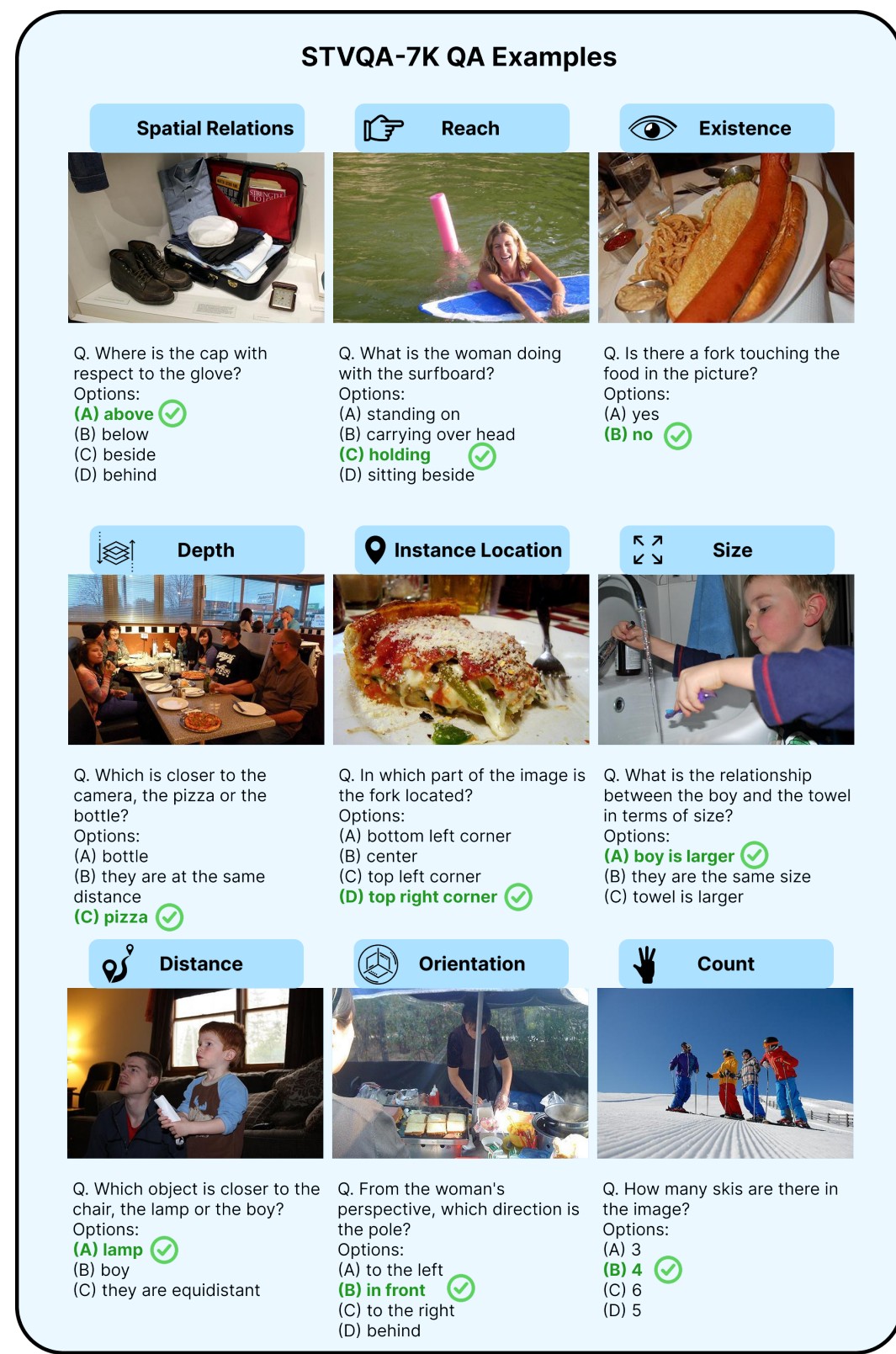

Figure 3: Examples of generated QA pairs across the nine spatial reasoning categories in STVQA-7K. Each category highlights distinct reasoning skills, ranging from relative spatial relations and depth ordering to distance, size, orientation, reach, location, count and existence.

across the nine spatial reasoning categories are shown in Figure 3, illustrating the diversity of question types in STVQA-7K.

**Scene Graph Adaptation.** Since each question focuses on specific objects and relationships within the broader scene, we derive question-aligned scene subgraphs that capture only the relevant spatial context. For each question, we extract content words through tokenization and lemmatization to obtain both singular and plural word forms. We then filter the original scene graph to retain only object nodes whose labels appear in the extracted question vocabulary. Relational triplets are preserved when both the subject and object entities are retained and the predicate appears in the question context. The resulting focused scene graph representations enable training the model to generate question-aligned region-of-interest subgraphs, encouraging it to localize attention, ground reasoning in relevant entities and relations, and ultimately learn where to focus within complex visual scenes.

# B  EXPERIMENTAL SETUP DETAILS

This section presents comprehensive evaluations of SPATIALTHINKER across multiple spatial reasoning benchmarks, demonstrating the effectiveness of our multi-objective dense reward design and data-efficient training approach.

## B.1  IMPLEMENTATION DETAILS

We build SPATIALTHINKER upon two strong open-source multimodal base models: Qwen2.5-VL-3B and Qwen2.5-VL-7B Bai et al. (2025), using them as backbones for policy optimization with reinforcement learning. No supervised fine-tuning is performed prior to RL training on our STVQA-7K dataset (Section 3.3). We employ GRPO Shao et al. (2024) as the advantage estimator as described in Section 3.2, using a rollout size of 8 samples per query and a sampling temperature of 1.0. The models are trained with a maximum context length of 16,384 tokens. The rollout batch size is set to 512, and the global batch size is 128. We train for 75 training steps i.e., 5 training episodes) on $4 \times$ NVIDIA H100 80GB GPUs. Training time totals around 13 hours for the 3B model and 15 hours for the 7B model.

The models are trained on high-resolution image inputs ranging from $512 \times 512$ to $2048 \times 2048$ pixels, to preserve fine-grained spatial information. All model parameters, including the vision encoder, are updated during training. We use the AdamW optimizer with `bf16` precision, a learning rate of $1 \times 10^{-6}$, and a weight decay of $1 \times 10^{-2}$. The KL penalty coefficient is set to $10^{-2}$. STVQA-7K is partitioned with a 90/10 train–validation split.

## B.2  EXPERIMENTAL SETUP

We evaluate SPATIALTHINKER across a diverse suite of 12 spatial understanding and real-world VQA benchmarks, covering both 2D and 3D understanding aspects to assess fine-grained spatial reasoning capabilities and real-world generalization. We compare against both proprietary and open-source baselines, including models specifically trained for spatial reasoning tasks. Our experiments address two key questions: (Q1) Does our spatial VQA data generation pipeline, combined with dense reward RL, improve MLLMs' general spatial reasoning capabilities? (Q2) How effectively can MLLMs learn spatial understanding from just 7K synthetic training samples, and how does this compare to models trained on orders-of-magnitude larger datasets?

**Benchmarks.** We evaluate models across six core spatial benchmarks, and six general-purpose VQA and real-world understanding datasets. The spatial benchmarks includes CV-Bench (Tong et al., 2024a) that measures 2D spatial relations, object counting, depth ordering, and distance reasoning. BLINK's Spatial Relations and Relative Depth tasks (Fu et al., 2024) test directional and positional understanding, and fine-grained point-level depth perception—particularly challenging as SPATIAL-THINKER receives no explicit point-level supervision during training 3DSRBench (Ma et al., 2024b) assesses egocentric 3D spatial reasoning via relational and multi-object comparisons. MMVP (Tong et al., 2024b) examines visual pattern recognition across attributes such as orientation, positional relations, existence, viewpoint, and size. SpatialBench (Cai et al., 2024) assesses general spatial comprehension across counting, existence, positional relationships, physical interactions such as reach, and size comparisons. Finally, SpatialReasonerEval (Ma et al., 2025a) emphasizes depth and

distance reasoning within 3D spatial tasks.

To assess broader generalization, we further evaluate models on six diverse real-world benchmarks. VStarBench (Wu & Xie, 2023) measures accurate localization and recognition of key objects in complex natural scenes. RealWorldQA (xAI, 2024) requires integrating visual inputs with common-sense and multi-step reasoning for real-world understanding. MME-RealWorld (Zhang et al., 2024) spans five challenging domains including optical character recognition in the wild, remote sensing, diagram and table interpretation, autonomous driving, and scene monitoring. RoboSpatial-Home (Song et al., 2025) simulates embodied spatial reasoning tasks involving object-object relationships, compatibility, and reference-frame switching (ego-centric, object-centric, and world-centric). We only use Configuration and Compatibility subsets of RoboSpatial-Home. MM-Star (Chen et al., 2024c) provides a holistic benchmark covering math, logical reasoning, instance recognition, and fine/coarse visual perception. HallusionBench (Guan et al., 2023) evaluates hallucination resistance in multimodal models, requiring accurate visual grounding to counteract entangled linguistic or perceptual illusions. Together, these benchmarks allow us to probe spatial and perceptual reasoning across synthetic, embodied, and naturalistic settings.

**Closed-Source MLLM Baselines.** Among proprietary models, we evaluate GPT-4o (GPT-4O-0513) (Hurst et al., 2024) and Claude 3.5 Sonnet (CLAUDE-3.5-SONNET-0620) (Anthropic, 2024), which represent the current state-of-the-art in commercial multimodal reasoning. These serve as upper bounds for spatial generalization under non-public training regimes.

**Open-Source Generalist MLLM Baselines.** We compare against generalist open-source MLLMs including Qwen2.5-VL 3B and 7B models (Bai et al., 2025), LLaVA-NeXT (Li et al., 2024b), Cambrian-1 (Tong et al., 2024a), and VLAA-Thinker (3B and 7B) (Chen et al., 2025a). These models represent state-of-the-art vision-language architectures, offering strong general visual reasoning but without specific spatial tuning.

**Open-Source Spatial MLLM Baselines.** We benchmark against specialized open-source models designed for spatial reasoning: SpaceLLaVA-13B AI & Mayorquin (2025a); Chen et al. (2024a) – a public re-implementation of SpatialVLM, SpatialRGPT-7B (Cheng et al., 2024) incorporates region-level supervision and explicit depth maps into training, RoboPoint-13B (Yuan et al., 2024), which instruction-tunes an MLLM to predict image key-point affordances for robotics and spatial affordance tasks, SpaceThinker (AI & Mayorquin, 2025c), a fine-tuned VLAA-Thinker model for spatial reasoning, and its improved successor SpaceOm (AI & Mayorquin, 2025b), which incorporates deeper chain-of-thought traces and Robo2VLM data (Chen et al., 2025b). Other baselines include SpatialReasoner (Ma et al., 2025a), trained with RL and explicit 3D representations, and SpatialBot (Cai et al., 2024), which integrates RGB and depth inputs for robust spatial perception.

In addition to the above, we compare against our training variants including supervised fine-tuning (SFT) baselines and vanilla GRPO trained with sparse rewards (accuracy and format only) to isolate the contribution of our dense spatial reward framework.

In addition to external baselines, we evaluate ablations on variants of our model trained with the STVQA-7K dataset: a supervised fine-tuning (SFT) baseline, and a sparse-reward RL baseline that optimizes only format and accuracy rewards, each weighted equally at 0.5. These ablations allow us to isolate the contribution of our proposed multi-objective dense spatial reward function.

**Evaluation Setting.** We report accuracy as the primary evaluation metric across all benchmarks. All models are evaluated under zero-shot settings, using greedy decoding (temperature = 0.0, max_new_tokens = 2048) to ensure deterministic and reproducible outputs. For models with specific reasoning templates such as VLAA-Thinker, SpaceThinker, and SpaceOm, we utilize their corresponding structured prompts. In line with their original training setup, SpatialRGPT receives depth inputs, while all other models are evaluated using RGB images alone. Our evaluation pipeline builds upon OpenVLThinker's evaluation framework (Deng et al., 2025), adapted to support our new benchmark and dataset formats.

### B.3  SPATIALTHINKER PROMPT FORMAT

We use a structured prompt to guide the model through a four-stage reasoning process, explicitly separated using the tags <observe>, <scene>, <think>, and <answer>. This format is enforced during training via a binary format reward $R_f \in \{0, 1\}$, with weight $w_{\text{format}} = 0.1$, which

verifies the presence, ordering, and validity of all required tags. The `<scene>` section must contain a JSON-encoded subgraph with object IDs, bounding boxes, and relational triplets, while the final answer must be clearly placed within the `<answer>` tags.

Each prompt also includes the input image dimensions in the form `Image size: {Width} × {Height}`, which are dynamically replaced with actual values. Including this information helps the model constrain predicted bounding box coordinates within image bounds, enabling better spatial localization. These coordinates are directly evaluated using IoU-based spatial rewards such as Complete IoU (CIoU), making dimension-aware prediction essential for optimizing structured spatial grounding.

> **SpatialThinker Prompt**
>
> You FIRST observe the image in <observe> </observe> tags, then visualise the relevant scene graph in <scene> </scene> tags, followed by thinking about the reasoning process as an internal monologue within <think> </think> tags and then provide the final answer. The final answer MUST BE put within <answer> </answer> tags, and only return the final choice including the correct option and answer within the answer tags, e.g., <answer> (C) The red cube is left of the green sphere </answer>.
> Image size: {Width} × {Height}

## B.4 DETAILS ON SFT TRAINING

To establish a comprehensive baseline for comparison with our reinforcement learning approach, we conduct supervised fine-tuning (SFT) experiments using the same base models (Qwen2.5-VL-3B and Qwen2.5-VL-7B) and training dataset (STVQA-7K). The SFT implementation utilizes LLaMA-Factory framework (Zheng et al., 2024) with Low-Rank Adaptation (LoRA) for parameter-efficient fine-tuning.

The training configuration employs LoRA with rank 8 applied to all available modules within the model architecture, enabling comprehensive adaptation while maintaining computational efficiency. Models are trained for 3 epochs totaling 645 training steps, using a context window length of 2048 tokens. We adopt BF16 mixed precision training with a learning rate of $1 \times 10^{-4}$, following a cosine learning rate schedule with a warmup ratio of 0.1.

For the SFT experiments, we train models directly on question-answer pairs without intermediate reasoning traces or chain-of-thought prompting. This design choice reflects the practical constraint that generating ground-truth reasoning traces would require additional dataset processing, annotation, and API credits budget. In contrast, reinforcement learning approaches with verifiable rewards (RLVR) naturally enables training with answer supervision alone, as the model learns to generate its own reasoning strategies through environmental feedback rather than imitating pre-specified reasoning patterns.

The SFT baseline serves a critical role in our experimental evaluation, providing direct evidence of the generalization advantages offered by reinforcement learning with dense spatial rewards compared to traditional supervised learning on the same dataset.

## B.5 DETAILS ON RL TRAINING

We implement reinforcement learning training using the EasyR1 framework (Zheng et al., 2025), building upon Qwen2.5-VL-3B and Qwen2.5-VL-7B as base models without any prior supervised fine-tuning. This direct application of RL to the base models enables us to isolate the effects of reward-driven learning from potential confounding factors introduced by intermediate training stages. Additionally, performing an SFT stage prior to RL would require generating ground-truth reasoning traces, which is limited by API budget. Moreover, explicit reasoning supervision is not strictly necessary—our multi-objective dense spatial rewards encourage the model to acquire structured reasoning and self-reflection abilities directly during RL training.

The training employs Group Relative Policy Optimization (GRPO) (Shao et al., 2024) as the advantage estimation method, configured with a rollout size of 8 samples per query at a sampling temperature

of 1.0. This configuration balances exploration diversity with computational efficiency, allowing the model to discover multiple reasoning strategies while maintaining stable convergence. The training process utilizes a rollout batch size of 512 and a global batch size of 128, processing data through 75 training steps (approximately 5 training episodes) to achieve convergence. The entire training pipeline runs on $4 \times$ NVIDIA H100 80GB GPUs, requiring approximately $\sim 13$ hours for the 3B model and $\sim 15$ hours for the 7B variant.

To preserve fine-grained spatial information critical for accurate object localization and spatial reasoning, models process high-resolution image inputs ranging from $512 \times 512$ to $2048 \times 2048$ pixels. The training configuration updates all model parameters including the vision encoder, enabling comprehensive adaptation to spatial reasoning tasks. Optimization employs AdamW with BF16 mixed precision, a conservative learning rate of $1 \times 10^{-6}$, and weight decay of $1 \times 10^{-2}$. The KL penalty coefficient is set to $10^{-2}$ to prevent excessive divergence from the base model distribution while allowing sufficient exploration for spatial reasoning strategies. The training utilizes a 90/10 train-validation split of the STVQA-7K dataset, with a maximum context length of 16,384 tokens to accommodate detailed scene descriptions and reasoning traces.

For baseline comparisons, we train vanilla GRPO models (Qwen2.5-VL-3B + Vanilla GRPO and Qwen2.5-VL-7B + Vanilla GRPO) using a simplified reward structure consisting solely of accuracy ($w_{acc} = 0.5$) and format rewards ($w_{format} = 0.5$), without the spatial grounding and count penalty components. This configuration represents standard RLVR approaches that rely on sparse final-answer supervision (DeepSeek-AI et al., 2025; Shen et al., 2025b; Chen et al., 2025a). The full multi-objective reward design employed for SPATIALTHINKER training, incorporating format, count, accuracy, and spatial rewards with lexicographic gating, is detailed in Section 3.1. The substantial performance improvements of SPATIALTHINKER over vanilla GRPO baselines demonstrate the critical importance of dense spatial supervision in teaching models to perform visually-grounded reasoning.

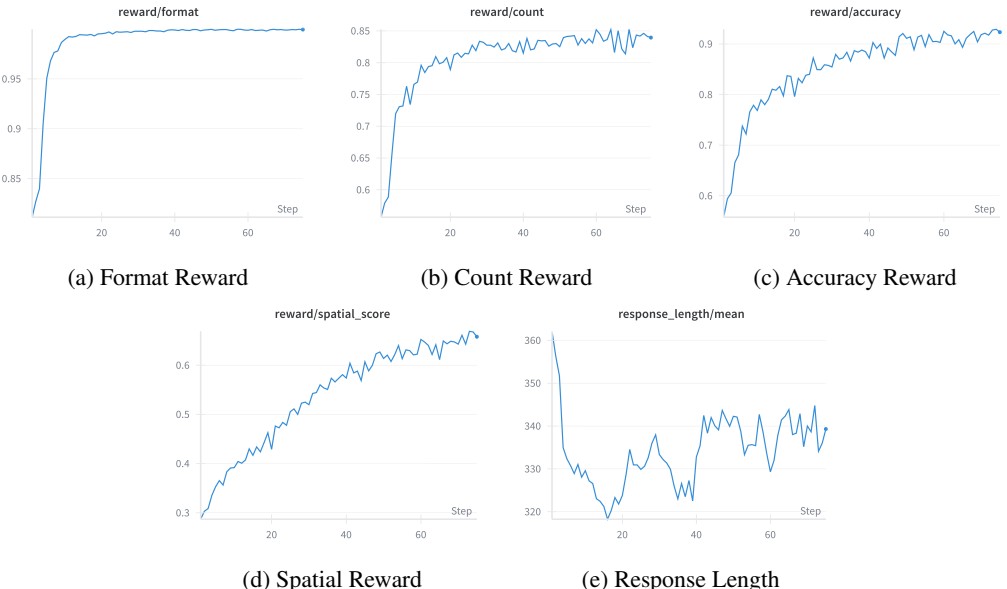

(a) Format Reward        (b) Count Reward        (c) Accuracy Reward

(d) Spatial Reward        (e) Response Length

Figure 4: RL training dynamics of SPATIALTHINKER. All reward components (a–d) improve consistently, reflecting stable optimization. Response length (e) shows a non-monotonic trend, indicating emergent reasoning strategies.

### B.5.1 SPATIALTHINKER RL TRAINING CURVES

Throughout reinforcement learning, all four reward components: format, accuracy, count, and spatial; demonstrate consistent and interpretable improvement, reflecting stable learning under our lexicographically gated, multi-objective reward structure. The format reward quickly converges early in training, indicating the model learns to produce structurally valid outputs that adhere to the required

scene-grounded reasoning format. Accuracy steadily improves across steps, highlighting the model's increasing ability to provide correct answers. Count reward rises consistently, showing that the model learns to focus on predicting only question-relevant objects and relations, rather than describing the entire scene. The spatial reward also improves gradually, indicating better object localization and grounding, as the model increasingly aligns predicted bounding boxes with ground truth annotations. Together, these trends reflect how each reward component scaffolds a different stage of the reasoning process, enforcing structure, correctness, focus, and grounding in tandem.

Response length initially declines, then rises again as it begins producing more deliberate, structured reasoning, signaling an "aha moment" where the model starts to produce more deliberate reasoning traces (DeepSeek-AI et al., 2025; Zhou et al., 2025). This emergent behavior suggests the development of internal problem-solving strategies, as the model learns to spend more "thinking time" before answering, consistent with the emergence of self-reflection and structured planning in its spatial reasoning process.

## C  REWARD DESIGN RATIONALE

Our reward design emerged from iterative refinement to address systematic reward hacking behaviors observed during training. Early experiments revealed that models readily exploit loopholes in reward functions—particularly when spatial localization rewards were provided without proper constraints. This section details our approach to designing a robust reward system that guides models toward genuine spatial reasoning while preventing degenerate solutions.

**Preventing Spatial Reward Hacking.**   Our initial reward formulation, which directly rewarded spatial localization quality, led to unexpected model behavior. Without constraints on generation quantity, models discovered they could maximize spatial rewards by generating numerous bounding boxes with varying coordinates. Through Hungarian matching that selects the best-matching boxes, even random predictions would occasionally yield high Complete IoU (CIoU) scores. This reward hacking manifested as models producing excessive, hallucinated objects while achieving poor task accuracy—the spatial reward was inflated despite the clutter of irrelevant predictions degrading actual performance. To address this exploitation, we introduced the Count Reward that penalizes deviations from expected object and relation counts. This reward serves dual purposes: (1) preventing reward hacking by constraining the generation space, and (2) encouraging models to focus on question-relevant scene elements rather than exhaustively describing the entire image. The count reward formulation provides a linear penalty proportional to relative deviations from ground truth counts, normalized to prevent domination by scenes with many objects.

**Scene Graph Filtering.**   Another form of overfitting emerged when training with complete Visual Genome scene graphs. Models would memorize exhaustive scene descriptions, including irrelevant background objects, leading to poor generalization. We addressed this by filtering ground truth scene graphs to retain only objects and relations relevant to the given question, focusing supervision on task-critical information.

**CIoU over IoU for Spatial Reward.**   For spatial localization, we adopt Complete IoU (CIoU) instead of standard IoU to compute the spatial reward. Unlike IoU, which returns zero when predicted and ground-truth boxes do not overlap, CIoU provides meaningful gradients by incorporating center distance, aspect ratio, and overlap (Zheng et al., 2020). This makes CIoU a denser and more robust supervisory signal during training.

**Balancing Supervision and Exploration.**   Our experiments reveal a crucial insight: models learn simple reward functions significantly faster than complex ones. Tasks with straightforward rewards (e.g., format compliance) show rapid improvements, while multi-component rewards require careful balancing. However, counterintuitively, highly detailed reward functions that attempt to supervise every aspect often degrade performance. Models overfit to maximize minute reward components, converging to template-style answers that score well on individual metrics while losing flexibility. We observed accuracy drops mid-training when rewards became too prescriptive, as models focused on reward optimization rather than genuine task understanding. Effective reinforcement learning requires providing guidance while preserving exploration space. Our final design addresses this by

providing soft signals through format checks, count constraints, and accuracy rewards, with spatial localization rewards activated only for correct answers. This maintains the delicate balance between guidance and exploration necessary for robust learning.

**Sequential Optimization via Lexicographic Gating.**    To prevent models from gaming individual reward components at the expense of task accuracy, we implement lexicographic gating (Skalse et al., 2022). Rewards are applied in a strict hierarchy: format $\succ$ {count, accuracy} $\succ$ spatial. This forces models to first master output formatting, then simultaneously learn to control generation scope and achieve correctness, before optimizing spatial grounding:

$$R_{\text{total}} = \mathbb{I}[R_{\text{format}} = 1] \cdot (w_{\text{format}} \cdot R_f + w_{\text{count}} \cdot R_c + w_{\text{accuracy}} \cdot R_a + \mathbb{I}[R_{\text{accuracy}} = 1] \cdot w_{\text{spatial}} \cdot R_s)$$

where $\mathbb{I}[\cdot]$ is the indicator function, with weights $w_{\text{format}} = 0.1$, $w_{\text{count}} = 0.2$, $w_{\text{accuracy}} = 0.5$, $w_{\text{spatial}} = 0.2$. This gated design ensures spatial rewards are only applied when the final answer is correct, aligning grounding quality with task success and preventing scenarios where models achieve high spatial scores through precise but irrelevant localizations.

## D    ABLATION ON DIVERGENCE CONSTRAINTS

Recent works such as DAPO (Yu et al., 2025; Vassoyan et al., 2025) argue that KL regularization can unnecessarily constrain policy updates and recommend removing the KL penalty entirely to allow freer exploration. In contrast, Huang et al. (2024) revisit divergence regularization and propose using a chi-squared penalty to better control overoptimization. Motivated by these findings, we ablate the effect of different divergence constraints in our reinforcement learning setup for spatial reasoning.

Table 6 reports results on CV-Bench 2D and 3D tasks (Tong et al., 2024a) for three variants of SPATIALTHINKER-3B: (i) no KL penalty, (ii) chi-squared divergence penalty with a coefficient of 0.01, and (iii) our default KL divergence penalty with a coefficient of 0.01. Removing the KL penalty leads to a noticeable drop in performance, particularly on 3D tasks. Using a chi-squared divergence penalty underperforms both the no-penalty and KL variants on several subtasks, especially depth and distance reasoning. The KL-regularized model achieves the best overall performance, yielding a CV-Bench average of 73.7% and providing the strongest results on 3D reasoning tasks.

These findings suggest that a modest KL penalty stabilizes policy updates and prevents reward overoptimization in our spatial reasoning setting, leading to more reliable improvements. While recent language-only alignment work has advocated for removing divergence constraints, our results indicate that retaining a small KL term remains beneficial for multimodal reasoning tasks where stability and coherent spatial grounding are crucial.

| Model Variant | Count | Relation | Depth | Distance | CV-Bench 2D | CV-Bench 3D | CV-Bench Avg. |
|---|---|---|---|---|---|---|---|
| SpatialThinker-3B + No KL Penalty | 65.5 | **76.8** | 74.8 | 70.2 | **71.2** | 72.5 | 71.9 |
| SpatialThinker-3B + Chi$^2$ (0.01) | 64.5 | 73.7 | 71.2 | 66.2 | 69.1 | 68.7 | 68.9 |
| **SpatialThinker-3B + KL (0.01)** | **68.5** | 73.5 | **79.7** | **72.8** | 71.0 | **76.3** | **73.7** |

Table 6: Ablation on divergence constraints for SPATIALTHINKER-3B on CV-Bench tasks. KL-regularization with $\beta = 0.01$ yields the highest overall average and strongest 3D reasoning performance.

## E    ADDITIONAL RESULTS: ABSTRACT REASONING

To further evaluate the generalization capacity of SPATIALTHINKER, we examine its performance on two abstract reasoning benchmarks: **Lego Puzzles** (Tang et al., 2025), which test compositional object reasoning and multi-step spatial reasoning, and **BLINK Multi-View** (Fu et al., 2024), which requires integrating spatial cues across multiple viewpoints, including visual-spatial understanding and perspective understanding. These tasks are not part of the training distribution and measure the ability of models to extrapolate structured reasoning skills to abstract domains.

| Model | Lego Puzzles | BLINK Multi-View |
|---|---|---|
| *Proprietary and Open-Source MLLMs* | | |
| GPT-4o | **57.7** | **54.1** |
| Claude 3.5 Sonnet | 53.6 | 51.9 |
| Qwen2.5-VL-3B | 29.9 | 42.9 |
| Qwen2.5-VL-7B | 35.8 | 44.4 |
| VLAA-Thinker-7B | 33.4 | 51.1 |
| SpaceThinker | 31.5 | 50.4 |
| SpaceOm | 32.0 | 48.9 |
| *Method Comparison (Trained on SpatialThinkerVQA)* | | |
| Qwen2.5-VL-3B + SFT | 34.7 | 42.1 |
| Qwen2.5-VL-3B + Vanilla GRPO | 27.0 | 45.9 |
| **SpatialThinker-3B (Ours)** | 33.9 | 45.1 |
| Qwen2.5-VL-7B + SFT | 36.6 | 44.4 |
| Qwen2.5-VL-7B + Vanilla GRPO | 29.7 | 51.9 |
| **SpatialThinker-7B (Ours)** | 37.7 | 52.6 |

Table 7: Results on abstract reasoning benchmarks. Lego Puzzles measure compositional reasoning over object arrangements, while BLINK Multi-View requires integrating multi-view spatial cues.

Across both tasks, SPATIALTHINKER-7B achieves the highest open-source performance improving over generalist and spatial MLLMs, and scoring 37.7% on Lego Puzzles and 52.6% on BLINK Multi-View, closely approaching GPT-4o and surpassing Claude 3.5 Sonnet on the latter. Interestingly, we observe that vanilla GRPO provides competitive performance on BLINK Multi-View but underperforms on Lego Puzzles, suggesting that dense spatial rewards offer complementary signals that better support compositional reasoning. These results demonstrate that the spatial grounding learned through reinforcement learning transfers to more abstract domains that require compositional and multi-view integration skills.

# F   DETAILED RESULTS: CV-BENCH

| Model | CV-Bench Tasks | | | | CV-Bench | | Avg. |
|---|---|---|---|---|---|---|---|
| | Count | Relation | Depth | Distance | 2D | 3D | |
| *Proprietary Models* | | | | | | | |
| GPT-4o | 65.9 | 85.7 | 87.8 | 78.2 | 75.8 | 83.0 | 79.4 |
| Gemini-1.5-Pro | 70.4 | 85.2 | 82.4 | 72.8 | 77.8 | 77.6 | 77.7 |
| Claude 3.7 Sonnet | - | 74.2 | 85.8 | 84.2 | - | 85.0 | - |
| *Open-Source General MLLMs* | | | | | | | |
| Qwen2-VL-2B | 54.7 | 22.6 | 16.7 | 31.7 | 38.7 | 24.2 | 31.5 |
| Qwen2.5-VL-3B | 61.5 | 58.3 | 67.3 | 53.0 | 59.9 | 60.2 | 60.1 |
| Qwen2.5-VL-7B | 55.9 | 82.2 | 70.0 | 66.0 | 69.1 | 68.0 | 68.6 |
| VLAA-Thinker-3B | 61.6 | 83.5 | 53.0 | 46.8 | 72.6 | 49.9 | 61.3 |
| VLAA-Thinker-7B | 47.0 | 74.6 | 61.3 | 59.2 | 60.8 | 60.3 | 60.6 |
| LLaVA-NeXT-34B | - | - | - | - | 73.0 | 74.8 | 73.9 |
| Mini-Gemini-HD-34B | - | - | - | - | 71.5 | 79.2 | 75.4 |
| Cambrian-1-34B | - | - | - | - | 74.0 | 79.7 | 76.9 |
| *Open-Source Spatial MLLMs* | | | | | | | |
| Spatial-LLaVA-7B | - | - | 57.3 | 52.2 | - | 54.8 | - |
| VisualThinker-R1-2B | 59.6 | 66.8 | 54.2 | 56.7 | 63.2 | 55.45 | 59.3 |
| Spatial-RGPT-7B w/ depth | - | - | 62.3 | 59.0 | - | 60.7 | - |
| RoboPoint-13B | - | 75.6 | 77.8 | 44.5 | - | 61.15 | - |
| SpaceThinker-3B | 61.0 | 69.2 | 70.5 | 61.3 | 65.1 | 65.9 | 65.5 |
| SpaceLLaVA-13B | - | 63.7 | 66.8 | 70.2 | - | 68.5 | - |
| SpatialBot-3B | - | 69.4 | 77.3 | 60.8 | - | 69.05 | - |
| *Method Comparison (Trained on STVQA-7K)* | | | | | | | |
| Qwen2.5-VL-3B + SFT | 30.2 | 77.5 | 61.2 | 75.5 | 53.9 | 68.4 | 61.2 |
| Qwen2.5-VL-3B + Vanilla GRPO | 67.5 | 73.7 | 64.0 | 69.2 | 70.6 | 66.6 | 68.6 |
| **SpatialThinker-3B (Ours)** | 68.5 | 73.5 | 79.7 | 72.8 | 71.0 | 76.3 | 73.7 |
| Qwen2.5-VL-7B + SFT | 33.3 | 78.9 | 64.8 | 77.7 | 56.1 | 71.3 | 63.7 |
| Qwen2.5-VL-7B + Vanilla GRPO | 58.9 | 78.8 | 79.3 | 73.7 | 68.9 | 76.5 | 72.7 |
| **SpatialThinker-7B (Ours)** | 68.7 | 86.7 | 81.2 | 76.2 | 77.7 | 78.7 | 78.2 |

Table 8: Detailed breakdown of CV-Bench (Tong et al., 2024a) results across Count, Relation, Depth, and Distance subtasks.

# G    DETAILED RESULTS: 3DSRBENCH

| Model | 3DSRBench Tasks | | | | Avg. |
|---|---|---|---|---|---|
| | Height | Location | Orientation | Multi-Object | |
| *Proprietary Models* | | | | | |
| GPT-4o | 53.2 | 59.6 | 21.6 | 39.0 | 44.3 |
| Claude 3.5 Sonnet | 53.5 | 63.1 | 31.4 | 41.3 | 48.2 |
| Gemini 2.0 Flash | 49.7 | 68.9 | 32.2 | 41.5 | 49.9 |
| Gemini 2.0 Flash (thinking) | 53.0 | 67.1 | 35.8 | 43.6 | 51.1 |
| *Open-Source MLLMs* | | | | | |
| Qwen2.5-VL-3B | 45.2 | 56.8 | 35.7 | 35.7 | 44.0 |
| Qwen2.5-VL-7B | 44.1 | 62.7 | 40.6 | 40.5 | 48.4 |
| Qwen2.5-VL-72B | 53.3 | 71.0 | 43.1 | 46.6 | 54.9 |
| Cambrian-1-8B | 23.2 | 53.9 | 35.9 | 41.9 | 42.2 |
| LLaVA-NeXT-8B | 50.6 | 59.9 | 36.1 | 43.4 | 48.4 |
| VLAA-Thinker-7B | 54.0 | 60.2 | 42.9 | 49.1 | 52.2 |
| *Open-Source Spatial MLLMs* | | | | | |
| SpatialBot-3B | 40.4 | 54.4 | 31.9 | 33.5 | 41.1 |
| SpaceLLaVA-13B | 49.3 | 54.4 | 27.6 | 35.4 | 42.0 |
| SpatialLLM-8B | 45.8 | 61.6 | 30.0 | 36.7 | 44.9 |
| SpatialRGPT-7B w/ depth | 55.9 | 60.0 | 34.2 | 42.3 | 48.4 |
| SpaceThinker-3B | 53.1 | 57.3 | 41.9 | 49.6 | 51.1 |
| *Method Comparison (Trained on STVQA-7K)* | | | | | |
| Qwen2.5-VL-3B + SFT | 51.1 | 58.3 | 42.7 | 48.1 | 50.8 |
| Qwen2.5-VL-3B + Vanilla GRPO | 48.9 | 57.9 | 42.5 | 47.2 | 50.1 |
| **SpatialThinker-3B (Ours)** | 52.6 | 61.8 | 43.4 | 49.8 | 52.9 |
| Qwen2.5-VL-7B + SFT | 50.6 | 66.3 | 43.8 | 47.9 | 53.6 |
| Qwen2.5-VL-7B + Vanilla GRPO | 54.3 | 64.7 | 45.5 | 50.4 | 54.7 |
| **SpatialThinker-7B (Ours)** | 52.0 | 70.3 | 45.5 | 50.9 | 56.4 |

Table 9: Detailed Breakdown of 3DSRBench (Ma et al., 2024b) Height, Location, Orientation, and Multi-Object tasks.

