# OpenReview forum: "SpatialThinker: Reinforcing 3D Reasoning in Multimodal LLMs via Spatial Rewards"
_ICLR.cc/2026/Conference — ICLR 2026 Conference Withdrawn Submission_

### Official Review · Reviewer_mfoQ · 2025-10-29

**Soundness:** 2
**Presentation:** 3
**Contribution:** 2
**Rating:** 4
**Confidence:** 2

**Summary:**

This paper presents SPATIALTHINKER, a 3D-aware MLLM designed for spatial reasoning tasks. The key idea is to combine structured scene graph grounding with multi-step reasoning via reinforcement learning. To facilitate this, the authors introduce a synthetic dataset, STVQA-7K, and a dense multi-objective reward function that guides the model to simulate human-like spatial perception and reasoning behavior.

**Strengths:**

(1)	The proposed reward function, which includes format, count, accuracy, and CIoU-based spatial rewards with lexicographic priority, provides a rich and structured learning signal that enables more stable and focused RL training.

(2)	SPATIALTHINKER is trained with only 7K samples but achieves strong generalization across 12 benchmarks, outperforming both supervised fine-tuning and sparse RL baselines.

(3)	The paper is the first to combine scene graph-based spatial grounding with online policy RL in a unified MLLM framework, allowing for structured spatial reasoning in a learnable and interpretable manner.

**Weaknesses:**

(1)	While the model introduces a dense multi-objective reward design, it remains unclear how each reward component individually contributes to training, as no detailed ablation is provided.

(2)	STVQA-7K focuses on single-image spatial reasoning, while most real-world spatial tasks require multi-view or video-based reasoning. It is unclear whether SPATIALTHINKER can handle such setting. Additionally, the scarcity of densely annotated spatial VQA data may limit its broader applicability.

(3)	The experiment section does not include evaluations on representative 3D spatial reasoning benchmarks such as VSI-Bench and SPAR-Bench. Given their relevance and popularity, readers are likely to be interested in how SPATIALTHINKER performs on these tasks.

**Questions:**

See Weaknesses

---

### Official Review · Reviewer_VHfg · 2025-10-31

**Soundness:** 3
**Presentation:** 3
**Contribution:** 2
**Rating:** 4
**Confidence:** 4

**Summary:**

This paper introduces SPATIALTHINKER, a 3D-aware multimodal large language model (MLLM) trained with reinforcement learning (RL). It integrates scene graph grounding and a multi-objective dense spatial reward. Unlike prior works that rely on massive 3D datasets , SPATIALTHINKER learns spatial reasoning through structured supervision and dense feedback signals from a compact synthetic dataset (STVQA-7K).

**Strengths:**

1.	The introduction of a dense, lexicographically gated reward combining format, count, accuracy, and spatial objectives is both elegant and effective. It provides continuous feedback for intermediate reasoning steps instead of only rewarding final answers , which is  a key step toward process-level RL in multimodal settings.
2.	Integrating question-focused subgraphs into the reasoning process is a well-motivated and interpretable approach. It encourages localized spatial grounding and generates more human-like reasoning behavior (observe → localize → reason → answer).
3.	Despite using only 7K samples, SPATIALTHINKER outperforms models trained with orders of magnitude more data (e.g., SpatialVLM 2B, SpatialRGPT 700K). The improvement over both SFT and vanilla GRPO baselines (∼1.8× higher gain) convincingly demonstrates the value of dense spatial rewards.

**Weaknesses:**

1.	Overall, the paper mainly contributes by introducing a reward-based framework that supervises dense spatial information through a scene-level reward, including counting and localization components. Although the dataset is generated using existing methods, the overall approach is relatively straightforward. The counting reward is a reasonable design, but the CIoU term is essentially a variant of the IoU metric, which has already been widely used in vision–language models trained with reinforcement learning.
2.	The paper lacks a detailed analysis of individual reward components, making it difficult to assess the specific contribution of each proposed reward. In particular, the experiments only compare three training settings — Qwen2.5-VL-7B + SFT, Qwen2.5-VL-7B + Vanilla GRPO, and SpatialThinker-7B (Ours) — without conducting the key ablation studies needed to separately evaluate the Dense RL framework and the Gating reward mechanism. As a result, the performance of each individual reward term remains unclear, and it is impossible to precisely measure their respective contributions. This raises two potential concerns: (1) In many spatial reasoning scenarios with a limited number of objects, the count reward might be easily overfitted or less informative, potentially reducing its overall influence. (2) The count reward weighting (set to 0.7/0.3) appears to be heuristic and lacks sensitivity analysis to verify robustness.
3.	The use of the <scene> tag encourages the model to generate scene-graph-related information and apply rewards for supervision. However, it is unclear whether this approach is actually better than simply guiding the model to describe scene relations or spatial layouts directly — that is, first encouraging perception-oriented generation (e.g., “describe the relative positions, attributes, or interactions among objects”) before reasoning. Such a design could potentially achieve a similar perception–reasoning separation without explicit reward supervision. Therefore, it remains uncertain whether the proposed scene-graph reward provides substantial advantages over more lightweight descriptive or instruction-based alternatives.
4.	The method relies on the Hungarian algorithm for matching predicted and ground-truth objects, which could become expensive when dealing with scenes that contain many objects. This may limit scalability to real-world datasets with dense spatial layouts. The authors might consider exploring more efficient or differentiable matching strategies, or providing an analysis of the algorithm’s computational overhead in such cases.

**Questions:**

1. How stable is the lexicographic gating under different hyperparameters (e.g., when accuracy gating is loosened)?
2. Could the authors share examples of visualization of output and some failure cases where the dense reward fails (e.g., overfitting to bounding box overlaps)?
3.Could the authors provide an analysis of how the accuracy of scene-graph prediction impacts the final results? In particular, does better scene prediction lead to more accurate reasoning or answers, and has this correlation been quantitatively evaluated?

---

### Official Review · Reviewer_zRGr · 2025-11-01

**Soundness:** 3
**Presentation:** 3
**Contribution:** 2
**Rating:** 4
**Confidence:** 3

**Summary:**

The paper introduces SpatialThinker, a 3D-aware multimodal large language model (MLLM) designed to improve spatial understanding in vision–language tasks. Unlike prior models that depend on large datasets, 3D inputs, or architecture changes, SpatialThinker uses RL with dense spatial rewards to learn human-like spatial reasoning. The model builds scene graphs of task-relevant objects and spatial relations, and reasons step-by-step to answer questions.

A key part of the work is STVQA-7K, a newly synthesized dataset of 7,000 spatial VQA samples used to train the model. The training involves a multi-objective reward framework that combines accuracy, spatial localization, and structured reasoning signals to encourage interpretable, grounded decision-making.

Results show that SPATIALTHINKER-7B, trained only on 7K samples, outperforms supervised fine-tuning, sparse-reward RL baselines, and even GPT-4o on multiple spatial reasoning and real-world VQA benchmarks. The findings suggest that integrating spatial grounding with dense reinforcement learning can enable efficient and robust 3D spatial reasoning in MLLMs without relying on massive data.

**Strengths:**

1. The lexicographically ordered multi-objective reward formulation is clearly defined and well-justified. The authors explicitly specify the gating logic, count penalty, and CIoU-based spatial localization terms, demonstrating a careful design that mitigates reward hacking and promotes interpretable policy learning.

2. The method achieves strong data efficiency, training on only ~7K samples while surpassing both supervised fine-tuning and sparse-reward RL baselines across six spatial reasoning and six real-world VQA benchmarks. The reported improvements are consistently summarized with reasonable margins.

3. The inclusion of divergence-constraint ablations (No-KL vs. Chi-square vs. KL) and per-reward component learning curves provides credible evidence of optimization stability and sound implementation.

**Weaknesses:**

1. The evaluation omits several strong and relevant baselines. Notably, it does not compare against recent closed-source models such as Gemini 2.5 or contemporary open-source spatial reasoning models such as SpatialReasoner and Space-Qwen, limiting the strength of the empirical claims.

2. Limited novelty in principle. While the integration of scene-graph grounding with gated multi-objective rewards is technically coherent, the individual components—scene-graph grounding and reinforcement learning with verifiable or structured rewards—are well-established. The contribution lies primarily in their combination, which is incremental relative to prior work on scene-graph-aware MLLMs and RLVR for spatial reasoning.

3. The term “dense reward” is used imprecisely. The proposed framework decomposes the total reward into multiple staged components rather than providing genuinely dense, continuous feedback throughout the trajectory; this distinction should be clarified to maintain conceptual rigor.

4. The ablation analysis remains incomplete. Key controls are missing, including the removal of the <scene> grounding stage (i.e., text-only CoT vs. grounded reasoning), a sensitivity study over reward weights, and a hybrid SFT→RL training variant commonly adopted in practice. These omissions make it difficult to isolate the contribution of each design choice.

5. The claim of “human-like perception-then-reasoning” is under-supported. The qualitative analyses are minimal, and the paper would benefit from detailed visualizations of subgraph predictions correlated with reasoning accuracy to substantiate the interpretability argument.

**Questions:**

Refer to Weakness

---

### Official Review · Reviewer_GKDu · 2025-11-01

**Soundness:** 3
**Presentation:** 3
**Contribution:** 2
**Rating:** 4
**Confidence:** 4

**Summary:**

The paper presents a framework that improves spatial understanding in VLMs through online RL. A data synthesis pipeline is introduced to generate the STVQA-7K spatial VQA dataset. GRPO training is further enhanced by adding dense spatial rewards, including 2D grounding and counting rewards. Experiments show that this training setup improves spatial understanding when data is limited, outperforming standard SFT and vanilla GRPO.

**Strengths:**

* The paper is well written and easy to follow.
* The experimental results are strong, showing consistent improvements over supervised fine-tuning and vanilla GRPO across multiple spatial reasoning benchmarks.
*  The method works well under limited data, which is practical and valuable for real-world settings.

**Weaknesses:**

* Although the results are good, the overall contribution feels somewhat limited. It is not surprising to see RL fine-tuning outperform supervised fine-tuning when data is limited. The grounding and counting rewards also seem broadly useful for visual understanding, not specifically tied to 3D reasoning.

* The data synthesis pipeline mainly relies on VG annotations and converting them into QA format through an LLM. This limits the scalability to the scope and patterns of VG and may not cover more complex or diverse spatial situations.

* The method claims to improve 3D spatial reasoning, but the reward signals are based on 2D pixel coordinates. This raises concern about whether the model is truly learning 3D spatial reasoning, or simply becoming more vision-centric and grounded in 2D layouts ( examples in Figure 1). It is also unclear why this would help answer orientation-related questions.

* The evaluation does not include 2D grounding benchmarks. Adding these results would help validate the contribution.
* After the RL training, if we do not prompt the model to produce the scene graph or follow the multi-step reasoning format, and instead ask a normal spatial question in a standard VQA style, how well does the model perform?

**Questions:**

Please see the weakness section.

---

### Note · Authors · 2025-11-14

I have read and agree with the venue's withdrawal policy on behalf of myself and my co-authors.